# Effects of Processing Conditions on the Properties of Monoammonium Phosphate Microcapsules with Melamine-Formaldehyde Resin Shell

**DOI:** 10.3390/polym15142991

**Published:** 2023-07-10

**Authors:** Shenjie Han, Jingpeng Li, Qingyun Ding, Jian Zang, Yulian Lu, Longfei Zhang, La Hu

**Affiliations:** 1Key Laboratory for Advanced Technology in Environmental Protection of Jiangsu Province, Yancheng Institute of Technology, Yancheng 224051, China; hansj@caf.ac.cn; 2Key Laboratory of High Efficient Processing of Bamboo of Zhejiang Province, China National Bamboo Research Center, Hangzhou 310012, China; 3College of Materials Science and Engineering, Yancheng Institute of Technology, Yancheng 224051, China; dqy13813028211@163.com (Q.D.); zangjian0104@163.com (J.Z.); 4School of Chemistry and Chemical Engineering, Yancheng Institute of Technology, Yancheng 224051, China; m17861905032@163.com; 5Key Laboratory of Wood Science and Technology of State Forestry Administration, Research Institute of Wood Industry, Chinese Academy of Forestry, Beijing 100091, China; zhanglongfei@caf.ac.cn; 6Guangxi Key Laboratory of Superior Timber Trees Resource Cultivation, Engineering Research Center of Masson Pine of State Forestry Administration, Guangxi Forestry Research Institute, Nanning 530002, China

**Keywords:** monoammonium phosphate, microcapsule, processing conditions, melamine-formaldehyde, in situ polymerization

## Abstract

To develop monoammonium phosphate (MAP) as a novel acid source for durable intumescent fire retardants (IFR), MAP microcapsules (MCMAPs) containing MAP as the internal core and melamine-formaldehyde (MF) as the external shell were prepared by in situ polymerization in this study. The influences of synthesis conditions (including reaction temperature, polymerization time, and reaction pH value) on the properties of obtained MCMAPs (MAP content, yield, morphologies, and thermal properties) were then investigated systematically. The morphologies, chemical structures, and thermal properties were characterized by optical microscopy, scanning electron microscopy (SEM), energy dispersive spectroscopy (EDS), Fourier transform infrared (FTIR) spectroscopy, and thermogravimetry analyzer (TGA). The results show that MAP was well encapsulated by MF resin. No microcapsules are obtained at <55 °C or with polymerization times <1 h. Optimal preparation conditions of reaction temperature, polymerization time, and reaction pH value are 75 °C, 3 h, and 5.5, respectively. Those results provide process reference and theoretical basis for preparing MCMAPs and could promote the application of MAP microcapsules in wood flame-retardant materials.

## 1. Introduction

Intumescent fire retardants (IFRs) are important environmentally friendly flame retardants present in a range of polymers and have received growing attention due to the desire for halogen-free fire retardants [1,2]. IFR systems usually consist of three active ingredients: an acid source, a carbon source, and a blowing agent [3]. However, conventional IFR systems formed by a simple mixture of these three components (e.g., ammonium polyphosphate, pentaerythritol, and melamine) are not durable due to their moisture sensitivity and weak affinity with polymer matrices [4]. Fortunately, microencapsulation can overcome these problems [5,6]. By forming an organic shell around the active ingredients, both the water resistance and polymer compatibility of the system were improved. Moreover, microencapsulation can effectively combine the three components of the IFR into a single particle <1 mm diameter, which can significantly enhance reactions between the ingredients upon heating [7,8,9].

In these microencapsulated IFR systems, the poorly water-soluble ammonium polyphosphate (APP) exhibiting a high degree of polymerization (>1000) is generally employed as both a core material and an acid source during microencapsulation [10]. For the shell material, melamine-formaldehyde (MF) resin is typically employed due to its favorable film forming ability, desirable mechanical properties, and suitability as a blowing agent [11]. The carbon source (e.g., pentaerythritol) can be introduced either by co-microencapsulation with the acid source or by incorporation into the MF shell [12,13,14]. In addition, APP with low degree of polymerization (DP = 50) has also been microencapsulated by MF resin to create an IFR system [15,16]. One main advantage of such systems is that no harmful solvents are employed during microencapsulation, as water or ethanol/water is generally used as the continuous phase. These microencapsulated IFR systems significantly improve the fire resistance of a range of polymers, such as polypropylene (PP) [17], rubber [18], ethylene vinyl acetate copolymer (EVA) [19], and polyurethane foam composites [20].

As a replacement for APP, monoammonium phosphate (MAP) and diammonium phosphate (DAP) have been employed as cheaper acid sources. For example, DAP microcapsules have been successfully prepared by interfacial polycondensation and coacervation techniques, but this required the usage of toluene as the continuous phase during encapsulation [21,22]. Furthermore, microencapsulation of DAP aqueous solution by MF resin has also been reported, but chloroform was used as the solvent to solubilize the poly(1,6-hexamethylene adipate glycol) (PHMA) carbon source [23,24,25]. In the absence of chloroform, irregularly shaped microcapsules were formed, which exhibited a rough surface and were prone to aggregation. Since MAP and DAP have high water solubility, their encapsulation is difficult without using harmful solvents. As such, facile and environmentally friendly methods for the preparation of microcapsules of water-soluble MAP for IFR systems are required.

To the best of our knowledge, few reports have been published into the encapsulation of MAP for an IFR system. Jaramillo et al. [26] adopted a solution-phase polymeric reaction between melamine and polyvinyl alcohol to obtain MAP microcapsules and investigate the effect of microencapsulation of MAP in the generation of fire-resistant coatings in the presence of tannins extracted from Pinus radiate. Ethanol was selected as both the dispersant for the MAP powders and the solvent for the MF pre-polymer. Importantly, the low solubility of MAP in ethanol (~1.22 g·L^−1^ at 25 °C) ensured successful microencapsulation. Thus, MAP microcapsules (MCMAPs) were prepared with MF shells to develop MAP as a novel acid source for durable IFRs. The effects of different conditions (reaction temperature, polymerization time, and reaction pH value) on the MAP content, yield, morphologies, and thermal properties of MCMAPs were investigated. The main objective of this article is to obtain the effects of different conditions (reaction temperature, polymerization time, and reaction pH value) on the MAP content, yield, morphologies, and thermal properties of MCMAPs. Through the above experimental characterization, the optimal experimental process will provide basic parameters for other related research.

## 2. Materials and Methods

### 2.1. Materials

Monoammonium phosphate (MAP), melamine, and sodium carbonate were provided by Tianjin Guangfu Fine Chemical Research Institute. Formaldehyde (37%) was purchased from Xilong Chemical Co. Ltd., Guangzhou, China. Absolute ethanol and acetic acid (36 wt%) were purchased from Beijing Chemical Works. Raw materials were of analytical grade and were used without further purification.

### 2.2. Preparation of the MF Prepolymer

A 37% solution of formaldehyde (40 g), melamine (20.7 g), and distilled water (28.1 g) was placed in a three-neck flask (250 mL) and mixed together. Then, the pH of the mixture was adjusted to 8.5–9.0 with sodium carbonate solution. The reaction mixture was then heated gradually to 90 °C using a heating bath (HBR 4, IKA^®^). The temperature was then maintained at 90 °C for 1.5 h when a transparent solution formed, after which time it was rapidly cooled using an ice/water bath. As the mixture solution was cooled to 40 °C, distilled water (88.8 g) was added to the mixture, and 20 wt% MF prepolymer was obtained.

### 2.3. Preparation of MCMAPs

MCMAPs were prepared according to the following in situ polymerization. Prior to microencapsulation, crystalline MAP was manually ground into a white powder and then passed through a 300-mesh sieve. The microencapsulation procedure of MCMAPs is shown in Figure 1. A portion of the MF pre-polymer (45 g) was placed into a three-neck flask (500 mL) and diluted with ethanol (135 g). The pH of the above mixture was adjusted to different values with acetic acid according to the Table 1. Then, 12 g MAP was added into the mixture. The agitation speed was then increased to 1000 rpm for 5 min in order to disperse MAP powder, after which time it was returned to 400 rpm. The resulting suspension was heated gradually to different polymerization temperature at a rate of 2.5 °C/min and reacted at different reaction time. The resulting MCMAPs were recovered by filtration and washed with distilled water three times. After drying at 50 °C in an air drying oven over 12 h, the MCMAPs were collected in sealed plastic bags for further characterization.

### 2.4. Characterization

#### 2.4.1. Yields and MAP Content of MCMAPs

The yields of MCMAPs were calculated using the ratio between the dry powder weight of MCMAPs and the raw material weight. In addition, ground samples (~0.5 g) were placed in a beaker with distilled water (80 mL). After stirring for 30 min using a magnetic stirrer, the resulting solution was allowed to stand overnight. The solid content of a sample of the supernatant liquid (50 mL) was estimated using an oven drying method (103 °C) and an analytical balance, and the MAP contents of MCMAPs were obtained by a simple conversion.

#### 2.4.2. Optical Micrographs of MAP and MCMAP 

Optical micrographs of MAP and MCMAP were obtained using an Olympus CX 31 optical microscope attached to a computer system. Samples were placed on glass slides, and a few drops of ethanol were added to disperse the powders. A cover glass was then placed on top of each suspension, and the micrographs were captured using a computer system.

#### 2.4.3. Field Emission Scanning Electron Microscopy (FE-SEM)

The surface morphologies of MAP and of MCMAPs were observed with FE-SEM (Hitachi S-4800, Hitachi, Tokyo, Japan) at an accelerating voltage of 10 kV. The particles were sprinkled onto double-sided tape and sputter-coated with a layer of gold. Energy dispersive spectroscopy (EDS) was used to analyze the elemental composition of the samples.

#### 2.4.4. Fourier Transform Infrared Spectroscopy (FTIR)

FTIR spectroscopy was carried out using a Nicolet Nexus 670 spectrometer between 4000 and 400 cm^−1^. Samples were mixed with KBr powder to prepare the KBr discs for analysis, and the resulting mixture was pressed to form a transparent tablet. 

#### 2.4.5. Thermogravimetric Analysis (TGA)

TGA was conducted under a flow of air on a DTG-60 series thermal analyzer (Shimadzu Science Instruments, Inc., Tokyo, Japan) between 40 and 550 °C at a heating rate of 5 °C/min. The sample weight employed was 2–3 mg.

## 3. Results and Discussion

### 3.1. Microencapsulation Mechanism

The in situ polymerization of microcapsules with an MF shell involves two key steps, namely pre-polymer preparation and shell formation [27,28]. The pre-polymer preparation step refers to the hydroxymethylation of melamine under alkalescent conditions [29]. Subsequent microencapsulation for MAP can then be divided into three stages, as shown in Figure 2. Under vigorous stirring, the MAP clusters are distributed in the MF pre-polymer/ethanol solution to obtain a suspension. At the same time, the MAP morphologies become smoother, which is attributed to MAP being soluble in water and slightly soluble in ethanol. Subsequently, the final reaction pH of the above suspension was adjusted to different values with acetic acid. The methylol groups of MF were influenced by hydrogen ions, and positively charged active MF pre-polymer was formed under acidic conditions. The active MF pre-polymers condensed with each other at different temperatures for different time to form small clusters according to Table 1, which gradually adhere to the MAP surface [27]. Finally, MAP had been surrounded by the MF clusters to obtain an integrated MF shell.

### 3.2. Morphology

Optical microscopy was used to understand the state of MAP in ethanol. The optical micrographs of MAP under different magnifications (see Figure 3) exhibit that the particles were well dispersed in ethanol. This suggests that ethanol, which has been used to scatter APP, could be a suitable harmless dispersant for MAP during microencapsulation [30]. Surprisingly, the obvious core/shell-like structure of MCMAP was also observed by optical microscopy (Figure 4), confirming the successful microencapsulation of MAP by the MF shell via in situ polymerization. Compared with the untreated MAP samples, some of the “core” MAP inside the microcapsules presented smoother profiles. This is likely due to the partial dissolution of MAP during shell formation, as observed previously in the preparation of APP-I microcapsules [15].

The SEM images of MAP and MCMAP (Sample 2) are presented in Figure 5. A number of the microcapsules exhibited smoother profiles than MAP, although the general morphologies were comparable. These results support the previous observations by optical microscopy. In addition, in the MCMAP group, particle aggregation was observed, which could be attributed to the agglomeration of small MAP or MCMAP particles during microencapsulation of MAP. Furthermore, few MF shells were observed, possibly resulting from incomplete encapsulation, which resulted in some MCMAP particles losing their MAP core material during the washing process [31]. Moreover, EDS analysis confirmed significant differences in the chemical compositions of MAP and MCMAP. For example, element C was observed only in the spectrum of MCMAP, while MCMAP contained lower elements P and O contents but a higher element N content compared to MAP. This confirmed the successful encapsulation of MAP by the MF shell.

### 3.3. Chemical Structure Analysis

The FTIR spectra of MAP, MF resin, and MCMAPs are demonstrated in Figure 6. To compare the chemical structures of the MCMAP series, a number of representative signals were examined. The absorption peaks of MAP were observed at approximately 3241 cm^−1^ (N-H asymmetric stretching vibration), 1445 cm^−1^ (N-H bending vibration), 1403 cm^−1^ (N-H bending vibration), 1288 cm^−1^ (P=O vibration), 1096 cm^−1^ (P-O symmetric stretching vibration), and 912 cm^−1^ (P-O symmetric stretching vibration) [26,32]. In the MF resin, absorptions at 1560, 1492, 1340, and 813 cm^−1^ were ascribed to the ring vibration of melamine from the MF shell, while the signal at 2972 cm^−1^ could be assigned to the stretching vibration of-CH_2_ groups within the resin [30,33]. In the MAP microcapsule spectra, absorptions at approximately 1558, 1504, 1340, and 813 cm^−1^ suggest that MF resin was present in the microcapsules. In addition, the broad bands at 1066 and 963 cm^−1^ may be attributed to the stretching vibrations of the phosphate groups (P=O and HO-P=O), corresponding to the spectral regions of 1096 and 912 cm^−1^ for MAP (See the blue and red boxes in the Figure 6) [25]. The majority of spectral characteristics for MAP were not observed in the MCMAP spectra, indicating that a coating of MF resin was present on the MAP surface.

### 3.4. Thermal Properties of MAP and MCMAP

TGA and DTG curves of MAP and MCMAP are shown in Figure 7. As observed for MAP microcapsules, the thermal properties of MCMAP were more closely related to those of MF resin than to those of MAP [25]. The weight loss of MAP can be divided into four steps. The first weight loss below 100 °C was assigned to the release of adsorbed water. The next two weight loss peaks at 189.03 and 376.09 °C ascribed to water and ammonia, respectively. In the last zone above 527.65 °C, MAP was transformed in polyphosphoric acid that dehydrates to phosphorus oxides [25]. However, the drop in mass reported at 300–350 °C for the MF resin was observed as a more gradual decline for MCMAP, likely due to interactions and bonding between the MF shell and the MAP core [15,25]. In addition, MCMAP exhibited initial signs of decomposition earlier than MAP due to the evaporation of water, methanol, and formaldehyde at lower temperatures [34]. However, the main decrease in mass for MCMAP was observed 136 °C higher than that for MAP. Furthermore, the final (char) residual mass of MCMAP at 550 °C was only 5.20%, suggesting that MCMAP was not a particularly effective IFR system due to its low MAP content. It was therefore expected that the thermal stability and char residual mass of MCMAP could be further improved by optimizing the operating parameters or by the incorporation of a carbon source (e.g., PVA, starch, or PEG) into the MF shell [35,36].

### 3.5. Effects of Operating Conditions on Properties of MCMAP

The operating conditions seriously affect the properties of MCMAP. It should be noted that no microcapsules were obtained from Samples 1, 5, and 6 (Table 1), indicating that formation of the MF shell is dependent on temperature, polymerization time, and reaction pH value [37]. At low temperatures (55 °C), the MF pre-polymer curing rate was too slow to yield shell formation [27], and with polymerization times <1 h, there was no sufficient time for forming integrated MF shells. Although white powders were obtained after filtration in these three experiments, no microcapsules were formed. Indeed, these powders were identified as water-soluble MAP, which dissolved rapidly in the washing process. Interestingly, Sample 8 was also successfully prepared in the absence of an acetic acid curing agent. It is expected that the reaction between MF pre-polymers was catalyzed by trace amounts of MAP dissolved in the water, which provided an acidic environment.

#### 3.5.1. Morphologies of MAP and MCMAP Samples

The SEM images of MAP and various MCMAP samples are shown in Figure 8. Upon increasing the reaction time from 2 to 3 h, no obvious difference in MCMAP morphology was observed. In contrast, temperature had a significant influence on MCMAP morphology. Upon increasing the temperature from 65 to 85 °C, the polymerization reaction between MF monomers became more rapid, leading to rougher MCMAP surfaces [27]. However, at 85 °C, a number of MF resin aggregates (red circles, Figure 8, sample 4) adhered to the surface of the MF membranes, thus indicating that polymerization at 85 °C was too rapid to allow effective microencapsulation.

In addition, the effect of reaction pH value was also examined in the preparation of MCMAP. There are no significant effects on MCMAP morphologies with decreasing the reaction pH value from 6.5 to 4.5. However, when the pH = 4.5, the yield is 25.7 ± 0.3%, but MAP content in MCMAPs is only 12.6 ± 0.1%, indicating that many of the products are MF polymers, which account for a large portion of the yield. Moreover, there are many MF polymers on the surface of MCMAPs as shown in Figure 8 (Sample 10). Increasing the pH value, the yield firstly increased and then decreased, and MAP content in MCMAPs increased. When the pH value was increased to 5.5, the yield was 19.0 ± 0.4%, but MAP content in MCMAPs was only 14.7 ± 0.1%, suggesting that many of the products were MCMAPs. As the pH increased above 5.5, the MAP content in MCMAPs was higher than the yield, meaning that many products were MAP, MF polymers were not polycondensed. When the pH value was too high, the catalytic effect was not obvious, and the polycondensation reaction was difficult to proceed, thus resulting in the yield and high MAP content in MCMAPs. On the contrary, the reaction rate of MF prepolymer was too fast, and self-polymerization occurred in the solution to form precipitates. Therefore, less MF polymer molecules were formed on the surface of MAP. When the pH value was suitable (pH = 5.5), the MF prepolymer diffused and was adsorbed on the surface of MAP, and the polycondensation reaction took place. The molecular weight of MF polymer increased continuously through the process of prepolymer–oligomer–polymer. Then, the water solubility of MF polymer also decreased, and the water-insoluble and cross-linked structure was formed. Finally, a film-wrapped core material (MAP) was formed to obtain MCMAPs [11].

#### 3.5.2. Thermal Properties of MCMAPs

Further in regard to the TGA results discussed previously for sample 2, the TGA curves for the MCMAP series are shown in Figure 9, indicating that in the initial pyrolytic stage (40–300 °C), the char residual masses of samples 3 and 9 were significantly higher than other samples. This indicates that the stability of the MF resin was mainly determined by temperature and reaction pH value during the poly-condensation step. Indeed, this is consistent with the fact that MF curing was controlled by reaction temperature and pH provided that the reactant molar ratios remained constant [33]. The optimal parameters of reaction temperature and reaction pH value in terms of the initial microcapsule thermal stability were 75 °C and 5.5, respectively. The reaction time had no influence on resin stability, since extension of the polymerization time did not affect the curing degree of MF.

In the subsequent main degradation process, a similar trend was observed for the whole MCMAP series up to the point where pyrolysis finished. However, the final char residual mass of the microcapsules (Figure 10) varied within the series (i.e., 4–15%). Indeed, temperature, curing agent dosage, and reaction time were all found to affect MCMAP char formation, with samples 3 and 9 exhibiting higher final values, likely due to the high stability of the MF resins in these samples, as described above. In addition, upon prolonging the reaction time from 2 h (sample 2) to 3 h (sample 7), the residual mass almost doubled due to the increase in MF shell materials in the microcapsules [38]. Indeed, the more stable the char IFR systems generated during degradation, the more effective their fire-retarding behavior should be. 

In conclusion, the effects of reaction temperature, polymerization time, and reaction pH on the MAP content, yield, morphology, and thermal properties of MCMAPs were investigated, and the results are as follows:(1)MCMAPs containing MAP as the core materials and MF as the shell materials were successfully fabricated by in situ polymerization.(2)Optical micrographs, FE-SEM, and SEM confirmed the successful microencapsulation of MAP.(3)The optimal reaction conditions for MCMAPs are reaction pH value 5.5, reaction temperature 75 °C, and polymerization time 3 h.

## 4. Conclusions

A series of MCMAPs obtained with MF resin as shell material and MAP as core material were fabricated under different processing conditions (reaction temperature, polymerization time, and reaction pH value) via in situ polymerization. The results of optical micrographs, SEM images, and EDS of MAP and MCMAP, FTIR, and TGA indicated that MAP was successfully microencapsulated by MF shell. By investigating the effects of operating conditions on the yield, MAP content in MCMAPs, morphology, and thermal properties of MCMAPs, the optimal preparation process was obtained with the reaction temperature 75 °C, polymerization time 3 h, and reaction pH value 5.5. The thermal stability of MCMAP was good, and the char residual mass was least under optimal synthesis process. This result may promote the development of MAP as a novel acid source for the preparation of durable IFRs.

## Figures and Tables

**Figure 1 polymers-15-02991-f001:**
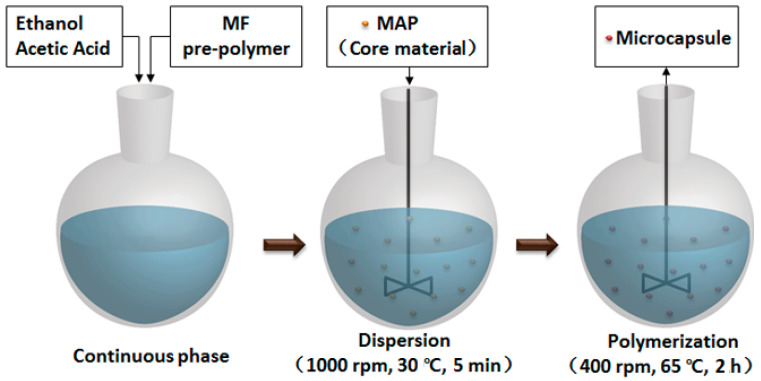
Schematic representation of the microencapsulation process.

**Figure 2 polymers-15-02991-f002:**
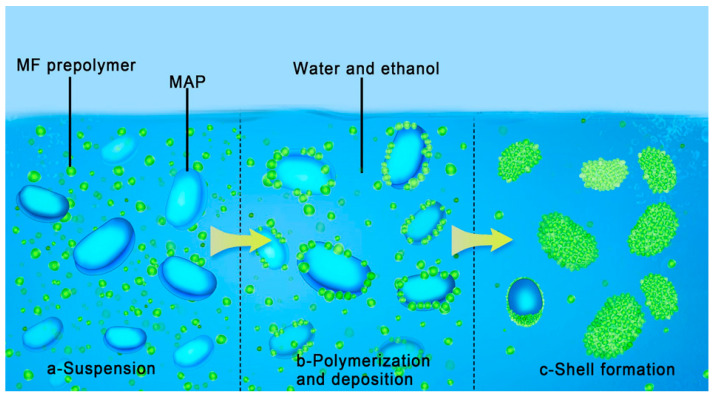
Illustration representing the in situ microencapsulation of MAP by MF resin.

**Figure 3 polymers-15-02991-f003:**
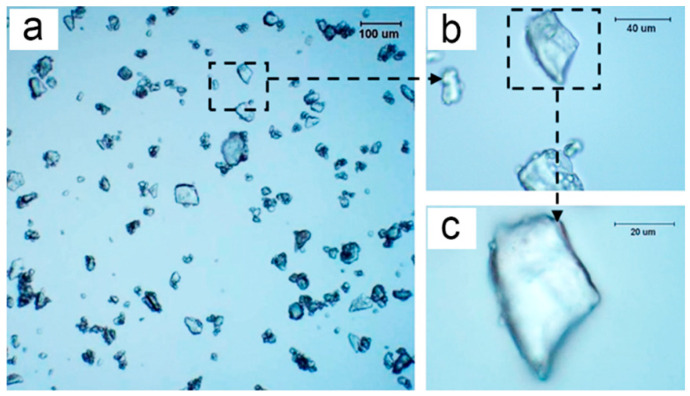
Optical microscopy images of MAP under different magnifications ((**b**) is a partial magnification of (**a**), (**c**) is a partial magnification of (**b**)).

**Figure 4 polymers-15-02991-f004:**
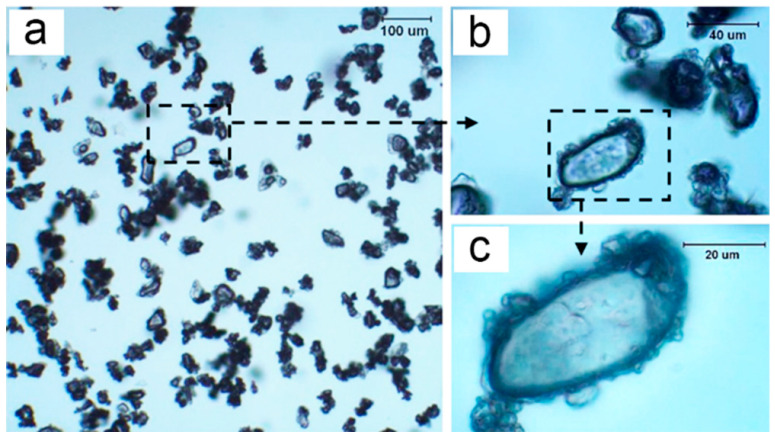
Optical microscopy images of Sample 2 under different magnifications, ((**b**) is a partial magnification of (**a**), (**c**) is a partial magnification of (**b**)).

**Figure 5 polymers-15-02991-f005:**
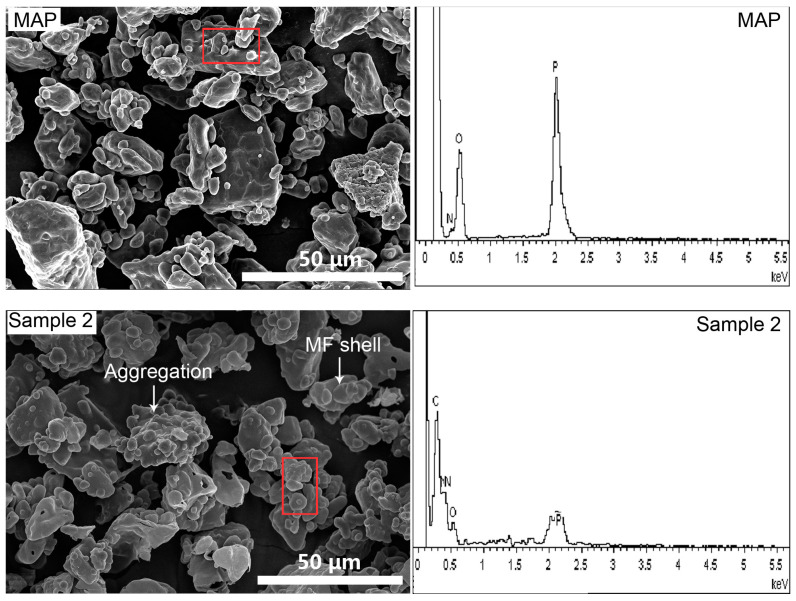
SEM images and EDS of MAP and MCMAP (red squares represent test areas for EDS analyses).

**Figure 6 polymers-15-02991-f006:**
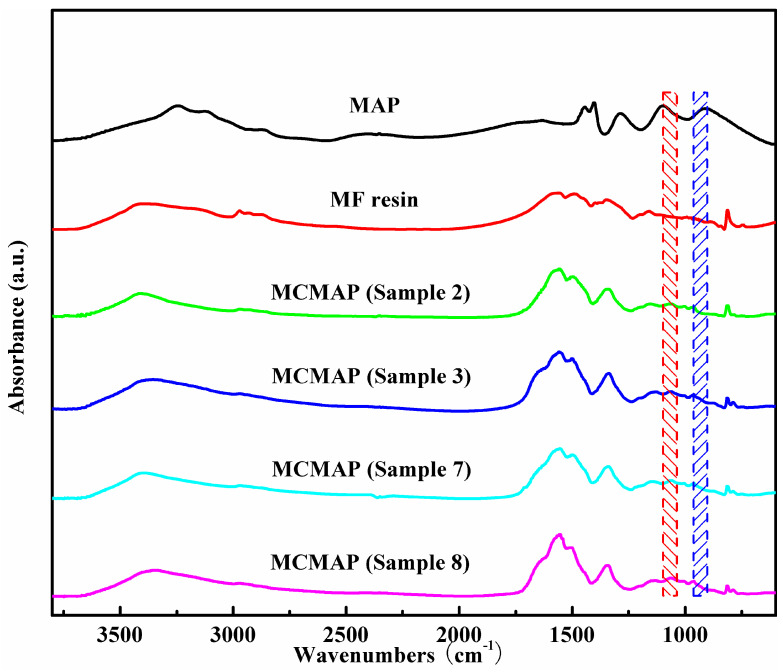
FTIR spectra of MAP, MF resin, and MCMAPs.

**Figure 7 polymers-15-02991-f007:**
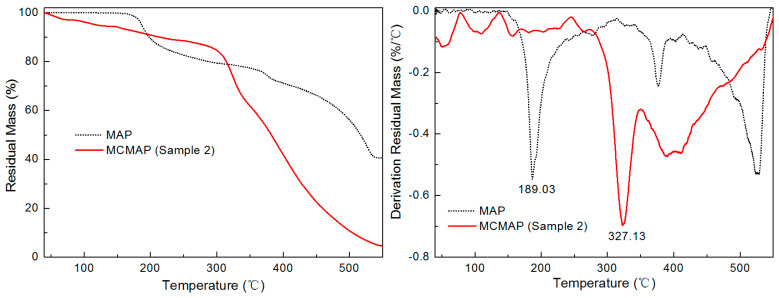
TGA (**left**) and DTG (**right**) curves of MAP and MCMAP (Sample 2).

**Figure 8 polymers-15-02991-f008:**
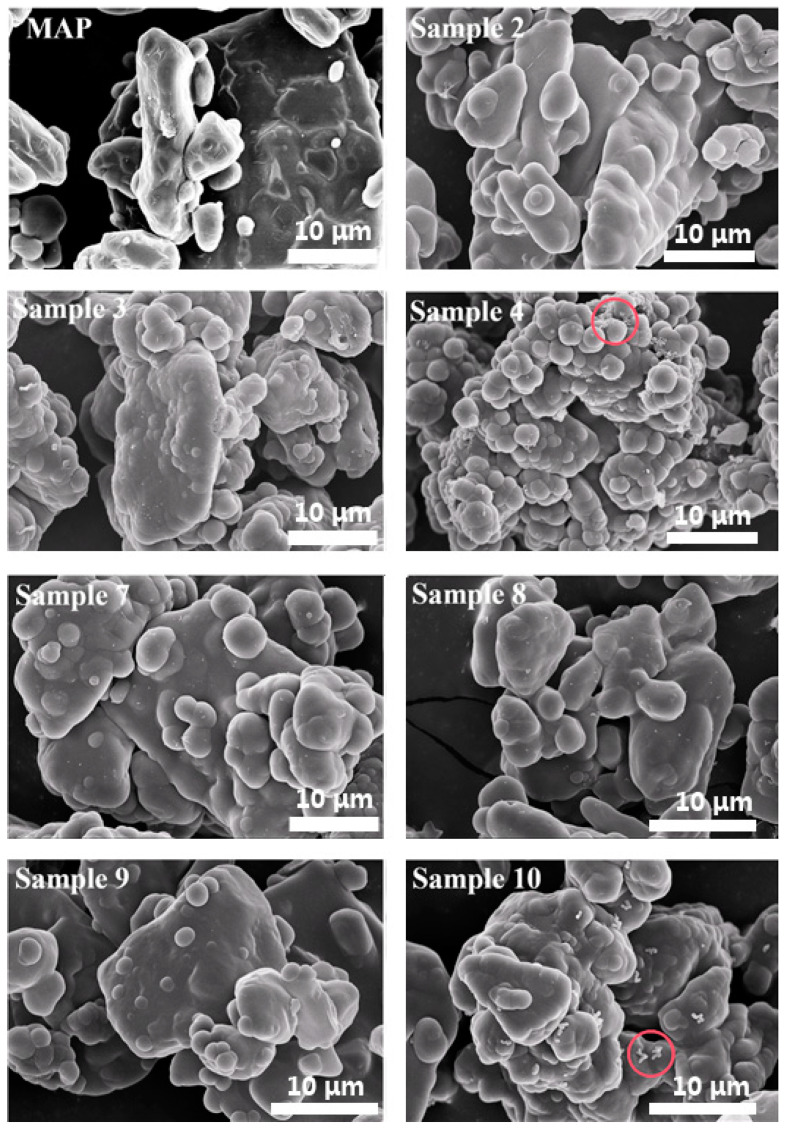
Effect of operating conditions on the MCMAP micromorphology (red circles in sample 4 was MF resin aggregates, and red circles in sample 10 was MF polymers).

**Figure 9 polymers-15-02991-f009:**
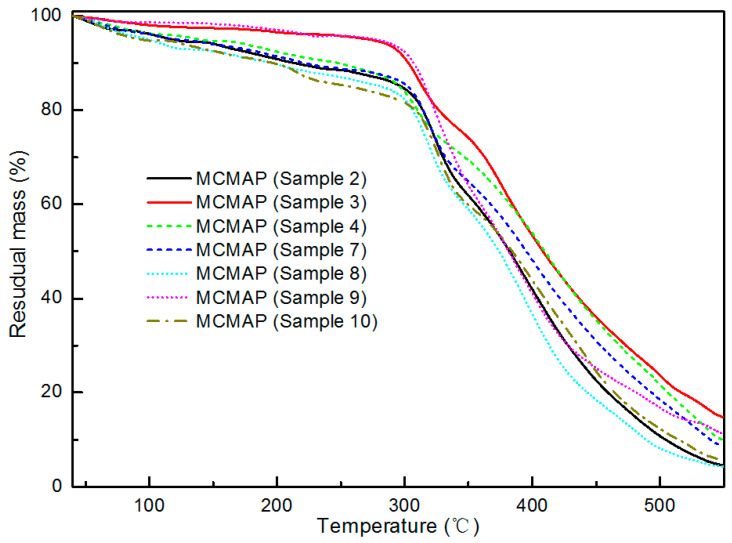
TGA curves of the MCMAPs.

**Figure 10 polymers-15-02991-f010:**
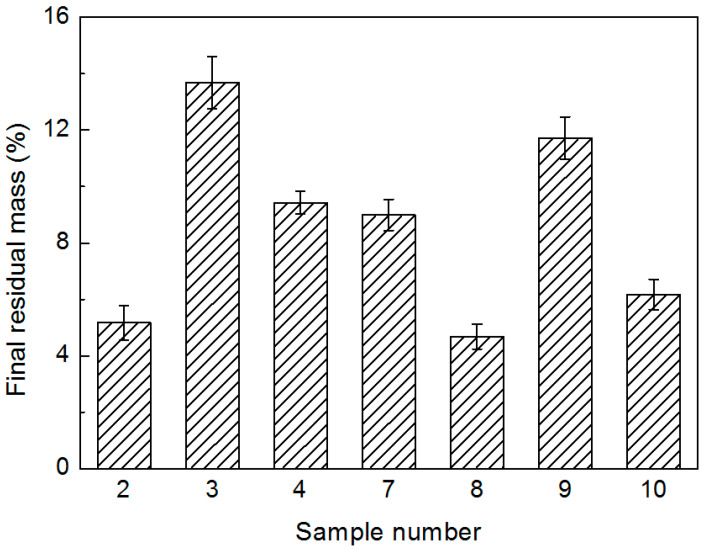
Final char residual masses of the MCMAPs.

**Table 1 polymers-15-02991-t001:** Reaction conditions and results for MCPAP preparation experiments.

Sample	Reaction Temperature (°C)	Polymerization Time (h)	Reaction pH Value	Yield (%)	MAP Content in MCMAPs (%)
1	55	2	5.0	0	None
2	65	2	5.0	21.7 ± 0.2	14.7 ± 0.1
3	75	2	5.0	22.5 ± 0.3	11.6 ± 0.2
4	85	2	5.0	23.8 ± 0.1	10.0 ± 0.2
5	75	0.5	5.0	0	None
6	75	1	5.0	0	None
7	75	3	5.0	26.1 ± 0.1	14.1 ± 0.2
8	75	3	6.5	15.3 ± 0.2	15.8 ± 0.4
9	75	3	5.5	19.0 ± 0.4	14.7 ± 0.1
10	75	3	4.5	25.7 ± 0.3	12.6 ± 0.1

## Data Availability

The data are available from the corresponding author upon reasonable request.

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
