# Peer review of "Effects of Processing Conditions on the Properties of Monoammonium Phosphate Microcapsules with Melamine-Formaldehyde Resin Shell"

_polymers, 2023, doi:10.3390/polym15142991_

Round 1

Reviewer 1 Report (Previous Reviewer 2)

Abstract

One last sentence is required. By the last end their should be another sentence to signify importance of the research result ??

For instance, add one sentence saying "Results showcased in this study recommends............................................."

========================================

Introduction

line 42:  "matrices[4]" is wrong, it should be "matrices [4]".  This mistake repeated many times in the manuscript.

line 81:  "Jaramillo et al." must be followed by ref directly

Last sentence in  the intro must be "Main objective of this research is to ............................................"

========================================

Results

It is clear that authors used ImageJ or other figure manipulation tool, would you mention that in details ?

Better than before, but still in need to editing / polishing.

Author Response

Abstract

One last sentence is required. By the last end their should be another sentence to signify importance of the research result ??

For instance, add one sentence saying "Results showcased in this study recommends............................................."

Response: The sentence of “Those results provide process reference and theoretical basis for preparing MCMAPs, and could promote the application of MAP microcapsules in wood flame retardant materials.” was acted as last sentence and added in the end of Abstract.

Introduction

line 42:  "matrices[4]" is wrong, it should be "matrices [4]".  This mistake repeated many times in the manuscript.

Response: We have corrected similar errors and checked the whole manuscript to avoid similar errors.

line 81:  "Jaramillo et al." must be followed by ref directly

Response: This error has been revised and the whole manuscript has been checked to avoid similar errors.

Last sentence in  the intro must be "Main objective of this research is to ............................................"

Response: The sentence “Main objective of this article is to obtain the effects of different conditions (reaction temperature, polymerization time, and reaction pH value) on the MAP content, yield, morphologies and thermal properties of MCMAPs. Through the above experimental characterization, the optimal experimental process will provide basic parameters for other related research.” was added in the end of Introduction.

Results

It is clear that authors used ImageJ or other figure manipulation tool, would you mention that in details ?

Response: We often measure sizes of particles by Nano-measure software (Version 1.2) instead of ImageJ. Other figure manipulation tools were Adobe Photoshop (PS), Microsoft PowerPoint (PPT), and Origin. The PS and PPT were used to combine images and add scales. The Origin was used to draw the FTIR spectra, TGA and DTG curves and final char residual masses of the MCMAPs. In addition, the particle sizes of MAP and MCMAPs were not measured in this paper to evaluate the effects of different conditions because MAP is slightly soluble in ethanol, and the sizes of MAP will change.

Reviewer 2 Report (New Reviewer)

The work addresses the interesting issue of preparing capsules with a flame retardant as the core and MF resin as the envelope. This is an interesting solution which could be successfully applied in the production process of, for example, chipboard or MDF. This is because by introducing such capsules into the bonding agent, usually in the form of urea-formaldehyde resin, it would be possible to ensure that they are evenly distributed throughout, resulting in a well-protected board after manufacture.

However, I do have some minor comments and would ask the Authors to respond to them:

1. Figure 6 is illegible and contains too much information. There is no point in describing all the peaks; it is best to focus only on the most relevant ones.

2. Chapter 3 in the title contains only Results. One can therefore get the impression that there is no discussion. Indeed, the discussion could be better and should be deepened. I propose having one paragraph at the end of this chapter where the most important research results are summarised and discussed.

Author Response

The work addresses the interesting issue of preparing capsules with a flame retardant as the core and MF resin as the envelope. This is an interesting solution which could be successfully applied in the production process of, for example, chipboard or MDF. This is because by introducing such capsules into the bonding agent, usually in the form of urea-formaldehyde resin, it would be possible to ensure that they are evenly distributed throughout, resulting in a well-protected board after manufacture.

However, I do have some minor comments and would ask the Authors to respond to them:

  1. Figure 6 is illegible and contains too much information. There is no point in describing all the peaks; it is best to focus only on the most relevant ones.

Response: Fig. 6 shows the characteristic peaks of MAP, MF-resin, and MCMAPs, respectively. The characteristic peaks of MAP and MF resin need to be given in detail, which is conducive to analyze the characteristic peaks of MCMAPs. The characteristic peaks of MCMAPs are somewhat biased, although all are given a little confusing, but it is conducive to readers to distinguish.

  1. Chapter 3 in the title contains only Results. One can therefore get the impression that there is no discussion. Indeed, the discussion could be better and should be deepened. I propose having one paragraph at the end of this chapter where the most important research results are summarised and discussed.

Response: The sentence “In conclusion, the effects of reaction temperature, polymerization time and reaction pH on the MAP content, yield, morphology and thermal properties of MCMAPs were investigated, the results are as follows: the optimal reaction conditions for MCMAPs are reaction pH value 5.5, reaction temperature 75℃ and polymerization time 3 h.” was added at the end of Chapter 3.

Round 2

Reviewer 2 Report (New Reviewer)

The discussion of the results is still missing and should be completed. Figure 6, as it is presented, is unacceptable to me. The authors should improve it and only give the ranges where changes are observed.

Author Response

The discussion of the results is still missing and should be completed. Figure 6, as it is presented, is unacceptable to me. The authors should improve it and only give the ranges where changes are observed.

Response: Chapter 3 has revised to “Results and discussion”. In fact, this section contains not only the results, but also the discussion. The discussion of the results has been deepened.

Figure 6 has been simplified and only given the ranges where changes are observed.

Round 3

Reviewer 2 Report (New Reviewer)

I have no further comments.

This manuscript is a resubmission of an earlier submission. The following is a list of the peer review reports and author responses from that submission.

Round 1

Reviewer 1 Report

In this paper, the author focuses on the MAP microcapsules by in situ polymerization in order to develop the new IFR. This paper contains interesting new results and is well worth publishing. However, there is some points I couldn’t understand. Therefore, I labeled this revision as minor. Detailed comments are provided below:

i)                   In page 11, the author writes the effect of reaction pH value. From SEM images, the author concludes that the reaction pH needs to be increased to obtain MCMAPs. Why is the pH 5.5 under optimal conditions? In addition, I would like to know the mechanism as to why these results are obtained for other conditions

ii)                 The author writes optimal preparation conditions are reaction temperature, polymerization time and reaction pH value are 75 C, 3h, and 5.5, respectively. How much better would the flame retardancy, etc. be if synthesized under those conditions?

Author Response

Review Report (Round 1)

In this paper, the author focuses on the MAP microcapsules by in situ polymerization in order to develop the new IFR. This paper contains interesting new results and is well worth publishing. However, there is some points I couldn’t understand. Therefore, I labeled this revision as minor. Detailed comments are provided below:

  1. i) In page 11, the author writes the effect of reaction pH value. From SEM images, the author concludes that the reaction pH needs to be increased to obtain MCMAPs. Why is the pH 5.5 under optimal conditions? In addition, I would like to know the mechanism as to why these results are obtained for other conditions.

Response 1: In the mixture system of MF prepolymer and MAP, the MF prepolymer was polycondensed to form a high molecular polymer under the action of a suitable acid catalyst. When the pH=4.5, the yield is 25.7±0.3%, but MAP content in MCMAPs is only 12.6±0.1%, indicating that many of the products are MF polymers, which account for a large portion of the yield. Moreover, there are many MF polymers on the surface of MCMAPs as shown in Fig.8 (Sample 10). Increasing the pH value, the yield firstly increased and then decreased, and MAP content in MCMAPs increased. When the pH value was increased to 5.5, the yield was 19.0±0.4%, but MAP content in MCMAPs was only 14.7±0.1%, suggesting that many of the products are MCMAPs. As the pH increased above 5.5, the MAP content in MCMAPs is high than the yield, meaning that many products are MAP, MF polymers are not polycondensed. When the pH value was too high, the catalytic effect was not obvious and the polycondensation reaction was difficult to proceed, thus resulting in the yield and high MAP content in MCMAPs. On the contrary, the reaction rate of MF prepolymer was too fast and self-polymerization occurred in the solution to form precipitates. Therefore, less MF polymer molecules were formed on the surface of MAP. When the pH value was suitable (pH=5.5), the MF prepolymer diffused and was adsorbed on the surface of MAP, and the polycondensation reaction took place. The molecular weight of MF polymer increased continuously through the process of prepolymer-oligomer-polymer. Then the water solubility of MF polymer also decreased and the water-insoluble and cross-linked structure was formed. Finally, a film-wrapped core material (MAP) was formed to obtain MCMAPs. (Han et al. Colloid. Surface. A. 2020).

  1. ii) The author writes optimal preparation conditions are reaction temperature, polymerization time and reaction pH value are 75 C, 3h, and 5.5, respectively. How much better would the flame retardancy, etc. be if synthesized under those conditions?

Response 2: When the polymerization time and reaction pH value were constant, the reaction temperature was variable, the yield was 22.5±0.3% and MAP content in MCMAPs was 11.6±0.2% at 75 oC reaction temperature. According to the report by Kage et al. (Kage et al., Adv. Powder Technol. 13 2002), the MF shell could be formed at reaction temperature lower than 75°C, but the formed MF shell was thin and incomplete. With the temperature rise, the polymerization reaction of MF was accelerated and the thickness of MF shell gradually increased and became denser. Increasing the polymerization time, keeping the reaction pH value (5.0) and reaction temperature constant (75 oC), the yield was 26.1±0.1% and MAP content in MCMAPs was 14.1±0.2% at 3h polymerization time. During the encapsulation process, diffusion, adsorption, uniform arrangement, and polycondensation requires a certain time. When reaction time prolonged, a part of the MF self-polymer in the reaction system was continuously adsorbed by MF wall, increasing MF shell materials in the microcapsules. When the reaction temperature was 75 oC and polymerization time was 3h, increasing the pH value to 5.5, the yield was 19.0±0.4%, and the MAP content in MCMAPs was14.7±0.1%.

Reviewer 2 Report

Introduction

Line 71  [24, 25].      modify this

Last paragraph in intro, there should be "Main objective of this article is..............................................."

Results

for Fig. 8, I do recommend using ImageJ tool to measure sized of particles and print down clearer scale bar

Conclusion

very week and miniatured, it should be clearer and longer. There shouldnt be copying from abstract to conclusion.

Language is ok

Author Response

Response to Reviewer 2 Comments

Review Report (Round 1)

Introduction

Line 71  [24, 25].      modify this

Response 1: Due to the loss of some references when applying the polymers-template, the references that were originally 27 and 28 are now 24 and 25. Now the missing references have been added back and the full text references have been carefully modified.

Last paragraph in intro, there should be "Main objective of this article is..............................................."

Response 2: Main objective of this article is to obtain the effects of different conditions (reaction temperature, polymerization time, and reaction pH value) on the MAP content, yield, morphologies and thermal properties of MCMAPs. Through the above experimental characterization, the optimal experimental process will provide basic parameters for other related research.

Results

for Fig. 8, I do recommend using ImageJ tool to measure sized of particles and print down clearer scale bar

Response 3: MAP is a colorless transparent tetragonal crystal. Prior to microencapsulation, crystalline MAP was manually ground into a white powder and then passed through a 300-mesh sieve, indicating that the size of MAP was less than 52 μm. In addition, MAP is slightly soluble in ethanol, and the sizes of MAP will change. ImageJ tool can measure the sizes of MAP, but whether the data are instructive is unclear. In this study, the sizes of MAP are not used as a measure of the success of microencapsulated MAP. Fig. 8 has been redrawn and scale bar was added.

Conclusion

very weak and miniatured, it should be clearer and longer. There shouldnt be copying from abstract to conclusion.

Response 4: We have checked the conclusion, and there's a difference between them, not copying and pasting.

The conclusion has been revised as following:

A series of MCMAPs obtained with MF resin as shell material and MAP as core material were fabricated under different processing conditions (reaction temperature, polymerization time, and reaction pH value) via in situ polymerization. The results of optical micrographs, SEM images and EDS of MAP and MCMAP, FTIR and TGA indicated that MAP was successfully microencapsulated by MF shell. By investigating the effects of operating conditions on the yield,MAP content in MCMAPs, morphology and thermal properties of MCMAPs, the optimal preparation process was obtained with the reaction temperature was 75 °C, polymerization time was 3h, and reaction pH value was 5.5. The thermal stability of MCMAP was good and the char residual mass was least under optimal synthesis process. This result may promote the development of MAP as a novel acid source for the preparation of durable IFRs.

Reviewer 3 Report

I am afraid I do not think that the article fits the scope of the special issue. I have a few comments to suggest authors improve their manuscript.

1-Line 81-82. These lines require references and must be explained in detail.

2-The novelty of the article is questionable if there are already a few articles.

3-The discussion is poor having only 9 references. It must be improved with more references. 

Moderate editing is required.

Author Response

Response to Reviewer 3 Comments

Review Report (Round 1)

I am afraid I do not think that the article fits the scope of the special issue. I have a few comments to suggest authors improve their manuscript.

Response: Ammonium polyphosphate (APP) is a well-known phosphorus flame retardant because of its low cost and excellent fire-retard capacity. Chen et al. (Chen et al. Journal of Central South University of Forestry & Technology, 2013.) treated wood panels with APP. THR and TSP were reported to be decreased by 43.11% and 54.56%, respectively, when compared with untreated samples. It indicates that APP has good effects on flame-retardation and smoke-suppression of woods. However, the APP IFR system is not durable because of its water solubility and poor compatibility with organic materials. Some of them can absorb moisture, and the penetration of water into the bulk of the wood promotes not only its decay (e.g. due to fungi) but also leads to mould growth, the corrosion of metal fixings and the reduction of mechanical strength. Therefore, APP in a polymer matrix can be easily hydrolyzed by water (moisture) and migrates to the surface leading to the worsening of flame retardancy and other polymer properties (Wang et al., Polym. Composite. 37 2016). Many investigations have been conducted to overcome these drawbacks, and various materials are successfully used for the microencapsulation of APP in recent years. This result may promote the development of MAP as a novel acid source for the preparation of durable IFRs. APP can effectively improve the flame retardancy of wood, which fits the scope of the special issue of wood flame retardancy.

1-Line 81-82. These lines require references and must be explained in detail.

Response 1: When we enter the search terms of monoammonium phosphate and encapsulation and intumescent fire retardants, very few articles are retrieved, and even fewer related articles. Jaramillo et al. obtained MAP microcapsules through a solution-phase polymeric reaction between melamine and polyvinyl alcohol and investigate the effect of microencapsulation of dihydrogen ammonium phosphate (MAP) in the generation of fire-resistant coatings in the presence of tannins extracted from Pinus radiate (Jaramillo et al. Coatings, 11 2021).

2-The novelty of the article is questionable if there are already a few articles.

Response 2: Although there have been many research papers on APP, there are few studies on MAP, which is widely used in wood. Two of the most effective fire retardants used in wood are DAP and MAP. The difference between APP and MAP is that APP has a greater solubility than MAP. MAP is a kind of inorganic salt, colorless transparent monoclinic crystal or white powder, easily soluble in water, insoluble in alcohol. Ammonium polyphosphate, also known as APP, is a mixture of ammonium orthophosphate and a variety of apps, easily soluble in water, solubility is greater than ammonium orthophosphate, but also can chelate metal cations, so that it can be retained in solution. Although there have been a few papers involving microencapsulated ammonium polyphosphate, there have been limited papers on MAP.

3-The discussion is poor having only 9 references. It must be improved with more references.

Response 3: More references are added in the discussion.
